# Human-in-the-Loop Optimization of Knee Exoskeleton Assistance for Minimizing User’s Metabolic and Muscular Effort

**DOI:** 10.3390/s24113305

**Published:** 2024-05-22

**Authors:** Sara Monteiro, Joana Figueiredo, Pedro Fonseca, J. Paulo Vilas-Boas, Cristina P. Santos

**Affiliations:** 1Center for MicroElectroMechanical Systems (CMEMS), University of Minho, 4800-058 Guimarães, Portugal; b13504@cmems.uminho.pt (S.M.); cristina@dei.uminho.pt (C.P.S.); 2LABBELS—Associate Laboratory, 4710-057 Braga, Portugal; 3LABBELS—Associate Laboratory, 4800-058 Guimarães, Portugal; 4Porto Biomechanics Laboratory (LABIOMEP), University of Porto, 4200-450 Porto, Portugal; pedro.labiomep@fade.up.pt (P.F.); jpvb@fade.up.pt (J.P.V.-B.); 5Centre of Research, Education, Innovation and Intervention in Sport (CIFI2D), Faculty of Sport, University of Porto, 4200-450 Porto, Portugal

**Keywords:** exoskeletons, human-in-the-loop control, metabolic cost estimation, work-related musculoskeletal disorders

## Abstract

Lower limb exoskeletons have the potential to mitigate work-related musculoskeletal disorders; however, they often lack user-oriented control strategies. Human-in-the-loop (HITL) controls adapt an exoskeleton’s assistance in real time, to optimize the user–exoskeleton interaction. This study presents a HITL control for a knee exoskeleton using a CMA-ES algorithm to minimize the users’ physical effort, a parameter innovatively evaluated using the interaction torque with the exoskeleton (a muscular effort indicator) and metabolic cost. This work innovates by estimating the user’s metabolic cost within the HITL control through a machine-learning model. The regression model estimated the metabolic cost, in real time, with a root mean squared error of 0.66 W/kg and mean absolute percentage error of 26% (*n* = 5), making faster (10 s) and less noisy estimations than a respirometer (K5, Cosmed). The HITL reduced the user’s metabolic cost by 7.3% and 5.9% compared to the zero-torque and no-device conditions, respectively, and reduced the interaction torque by 32.3% compared to a zero-torque control (*n* = 1). The developed HITL control surpassed a non-exoskeleton and zero-torque condition regarding the user’s physical effort, even for a task such as slow walking. Furthermore, the user-specific control had a lower metabolic cost than the non-user-specific assistance. This proof-of-concept demonstrated the potential of HITL controls in assisted walking.

## 1. Introduction

Musculoskeletal disorders currently stand as the most prevalent work-related health problem among European workers. A 2019 report commissioned by the European Agency for Safety and Health at Work disclosed that 60% of workers with work-related health issues reported suffering from a musculoskeletal disorder, and 30% experienced muscular pain in the lower limbs [1]. Activities such as carrying or lifting heavy loads and working in uncomfortable positions, such as standing or squatting, pose a higher risk of lower limb musculoskeletal disorders [1,2,3]. Furthermore, it has been estimated that 63% of workers who carry or lift heavy loads exhibit musculoskeletal disorders [1].

Lower limb exoskeletons are starting to gain notoriety among industry employers as devices able to reduce musculoskeletal stress and metabolic cost by augmenting, assisting, or reinforcing the workers’ musculoskeletal abilities [4,5,6]. Promising results indicated that active lower limb exoskeletons can reduce the acute physical stress and strain of workers during tasks like walking and carrying heavy loads [6,7]. However, the adoption of these devices is still limited due to usability concerns and a lack of personalized assistance. Typically, an exoskeleton’s assistance is adjusted by performing an offline optimization, and the inter-subject variability in users’ motor needs and responses to devices is disregarded [8,9]. This raises the need for employing adaptive control strategies that adjust the assistance to each user individually and in real time.

In the last few years, human-in-the-loop (HITL) controls have been implemented in exoskeletons, to automatically adjust the control parameters (such as the assistive torque) according to the physiological measurements of each user [9]. The two most common optimization algorithms used for this purpose are the CMA-ES [9,10,11] and Bayesian optimizer [12,13,14]. Overall, the HITL controls available in the literature have proven to be successful in optimizing the users’ metabolic cost [8,9,10,11], muscle activity [15,16], and gait speed [17]. For instance, Zhang et al. [9] developed a HITL control for a tethered ankle exoskeleton and obtained average metabolic cost reductions of 24.2% and 14% compared to walking in a zero-torque condition and without the device, respectively. Ding et al. [12] and Xu et al. [15] developed a HITL control for hip exoskeletons capable of reducing the participant’s metabolic cost and muscle activity by 17.4% and 13%, respectively, when compared to walking without the device. Furthermore, Slade et al. [10] optimized the assistance of a wearable ankle exoskeleton, obtaining a 17% reduction in the users’ metabolic cost and a 9% increase in their selected speed, compared to walking without the exoskeleton. Han et al. [16] reduced the muscle activity of one participant wearing a tethered ankle exoskeleton by 31.6%, 26.2%, and 10.7% for normal walking, uphill walking, and loaded walking with 5 kg, respectively, compared to walking with a zero-torque condition. Therefore, HITL controls have the potential to be used in industrial exoskeletons to lower workers’ metabolic cost while carrying and lifting heavy loads.

Indirect calorimetry is the standard metabolic cost estimation method [18]. It requires the use of a respirometer device to measure the flow of oxygen and carbon dioxide through a person’s respiratory system [18]. Despite obtaining reliable metabolic cost estimations, this method requires expensive equipment [19], takes a long time (ranging from 2 to 3 min) to obtain a steady-state metabolic cost [20], and generates a noisy signal [21]. Furthermore, the obstructive design of the respirometer device makes it impractical for workers to wear it in real industrial environments [22].

An alternative to indirect calorimetry is the use of machine- [18,22,23] or deep-learning [24,25] regression models that estimate metabolic cost based on data acquired from wearable and non-intrusive sensors. For instance, Slade et al. [22] used a linear regressor that received inertial measurements from the shank and thigh as inputs, obtaining a mean absolute percentage error (MAPE) of 13.7%. Lucena et al. [19] developed a hierarchical linear regressor, with multiple inputs— wrist, ankle, and hip acceleration, and heart rate—and obtained a root mean squared error (RMSE) of 0.613 kcal/min. More complex machine-learning models have been used, such as a support vector machine that presented an RMSE of 0.78 kcal/min only relying on foot pressure and acceleration as inputs [23]. Ingraham et al. [18] also employed a linear regressor while studying the use of various input signals, namely the acceleration, heart rate, respiratory rate, electrodermal activity, and skin and ambient temperatures. Despite being considered a simple model, the linear regressor of Ingraham et al. [18] obtained an RMSE of 1.03 W/kg. Lopes et al. [24] developed a convolutional neural network that estimated metabolic cost based on feet, leg, pelvis, and torso acceleration; leg muscle activity; and heart rate, achieving an RMSE of 0.36 W/kg. Overall, linear regressors or some variation of these have been the most used regression models [18,19,22]. Additionally, the input signals that resulted in better metabolic cost estimation are (i) waist, wrist, and ankle acceleration; (ii) muscle activity; (iii) heart rate; (iv) breath frequency; and (v) minute ventilation [18,19].

To the authors’ best knowledge, the recent work of Slade et al. [22] is the only study that performed a real-time experiment using a regression model to estimate metabolic cost, achieving a MAPE of 23%. Additionally, several regression models for metabolic cost estimation have been developed in the literature; however, these models have not yet been integrated into HITL controls. This work proposes a HITL control designed for an industrial exoskeleton, intending to assist workers in reducing their physical effort while walking, with the potential to be extended to load-carrying and -lifting assistance. In this manuscript, our goal is to present a HITL control capable of performing an online and user-specific optimization of an exoskeleton’s assistive torque. This optimization aims to reduce the physical effort experienced by a knee exoskeleton user during walking. The users’ physical effort is evaluated using their metabolic cost and their interaction torque with the exoskeleton, to provide a more comprehensive understanding of human effort. This assessment of physical effort is in line with a definition of physical activity that is any human movement resulting from a muscular force that causes an energy expenditure [26]. As far as we are aware, this is the first study that combines both a metabolic parameter and a muscle activity parameter in the cost function of a HITL control. Furthermore, this work innovates by using an inertial data-based regression model to estimate, in real time, metabolic cost, instead of relying on the indirect calorimetry method, as previously employed in the literature on HITL controls [9,11,21]. Our work contributes to scientific knowledge by demonstrating that regression models can be used to estimate physiological signals, namely metabolic cost, in real time; thus, opening a new avenue for HITL control applications in industrial exoskeletons.

## 2. Materials and Methods

This section presents the development and validation of the HITL control designed for assisted walking. Figure 1 depicts an overview of all stages of this control. The user is equipped with a smart assistive device named SmartOs. This includes an active knee exoskeleton and a team-owned wearable inertial sensor system [27]. The chest, wrist, waist, and ankle acceleration and the user’s body mass index (BMI) are used as inputs to an exponential Gaussian process regressor (EGPR) model, to estimate the metabolic cost. The user’s physical effort information (i.e., estimated metabolic cost and human–exoskeleton interaction torque) is subsequently sent to a covariance matrix adaptation evolutionary strategy (CMA-ES) optimization algorithm. This iteratively tests various combinations of control parameters to find the flexion and extension torque peaks that minimize the cost function (i.e., the user’s physical effort). Lastly, a torque-tracking control is used to generate the torque profile for each set of control parameters and to ensure that the exoskeleton actuator’s torque replicates the desired pattern. A detailed description of each stage is presented below.

### 2.1. SmartOs System

SmartOs is a modular, smart, and fully wearable assistive exoskeleton for the lower limbs, namely the knee and/or ankle joints of the H2 exoskeleton (Technaid, Madrid, Spain). It includes a central controller unit (CCU, UDOO X86 with an Intel^®^ Celeron N3160 up to 2.24 GHz processor) responsible for managing the communication between all system modules and for running motion analysis tools and high-level controllers at 100 Hz. The CCU interfaces with two development boards with lower computational capabilities (STM32F4-Discovery board, STMicroelectronics, Geneva, Switzerland). These boards consist of (i) a low-level board, which manages the mid- and low-level controllers at 1 kHz and interfaces with the active knee and ankle exoskeletons (H2 exoskeleton, Technaid, Spain) through a control area network protocol; and (ii) the wearable motion LAB board, handling real-time data acquisition of team-developed sensor systems (such as the InertialLab system [27]). Further details are presented in [28].

Within the scope of this study, we used the knee exoskeleton for the right lower limb (as illustrated in Figure 1). It consists of an electrical actuator (EC60 100W Flat Brushless (Maxon, Bad Homburg, Germany)) coupled to a gearbox (CSD-20-160-2A-GR (Harmonic Drive, Tokyo, Japan)), with one degree of freedom in the sagittal plane and a gear ratio of 160:1, capable of providing a nominal torque of 20 Nm. The device is also equipped with various sensors: (i) a potentiometer that measures the knee’s angle; (ii) strain gauges that measure the user–exoskeleton interaction torque; and (iii) Hall effect sensors that assess the motor’s torque. The power supply system consists of a lithium iron phosphate battery (LifePO4) with a voltage of 22.4 V and a capacity of 12 Ah. The exoskeleton is capable of assisting gait speeds from 0.5 to 1.6 km/h. The device is tightly secured to each user’s leg using four adjustable straps, two at the shank and two at thigh level. The straps allow adjusting the device to different users’ leg widths. Moreover, the mechanical structure of the device allows adjustment to user heights from 1.5 to 1.95 m and body masses from 50 to 100 kg.

Moreover, we used four inertial measurement units (IMUs) from a team-owned wearable inertial sensor system [27] to measure the 3D acceleration signals of the chest, right wrist, left waist, and right ankle at 100 Hz. Figure 1 presents the on-body IMU locations in blue.

### 2.2. Metabolic Cost Estimation

The metabolic cost is estimated in real time by a machine-learning regression model (blue block of Figure 1), namely an EGPR. From our previous benchmark analysis [29], we verified that the EGPR was the best-performing model when compared to other machine and deep learning regression models, such as a boosted decision tree, a bagged decision tree, a support vector machine, and a convolutional neural network [25]. The EGPR model, with an optimized Sigma of 0.15, was trained and validated offline with data from a publicly available dataset with a sample size of 10 participants [18], through the leave-one-subject-out cross-validation method. The data selected from the dataset included acceleration measurements taken at the chest, right wrist, left waist, and right ankle, and the BMI of 10 participants. These input signals were selected since they presented better results than the electrodermal activity and the acceleration derivative and vector norm in a previous study [29]. Additionally, only the data from relevant industrial activities, namely the standing, sitting, and walking activities, were selected. The ground-truth metabolic cost was estimated by indirect calorimetry.

We implemented the trained regression model in a SmartOs’ CCU board (C++, Ubuntu mate) to estimate the metabolic cost in real time every 10 s. This time window guaranteed at least two respirations and was considered adequate to estimate the instantaneous metabolic cost [29]. For this purpose, the system executes the following steps in real time: (i) measuring the 3D acceleration signals of the four mentioned on-body locations; (ii) filtering these signals with a 4th-order Butterworth low-pass filter at 20 Hz [22,24,25], which offers a good trade-off between signal attenuation and preservation of relevant characteristics; (iii) reorganizing the data into 10-second windows [30]; (iv) creating the feature vector by computing the mean absolute deviation (MAD) of each variable [31]; (v) normalizing each feature using the z-score method; and (vi) using the previously trained EGPR regressor to estimate the metabolic cost based on the input features. Figure 2 presents the sequence of the pre-processing methods, which was established based on the results of our previous study [29].

### 2.3. HITL Optimization

The HITL optimizer is responsible for adjusting two exoskeleton control parameters over various iterations to find the set of parameters that minimize the user’s physical effort (the cost function). This effort is measured by a weighted sum of the user metabolic cost estimated by the EGPR model and the human–exoskeleton interaction torque, with equal weights for both parameters. The interaction torque has a direct relation to the user’s participation and muscle effort, since the muscular activity decreases as the users reduce their interaction torque. Combining physiological measurements that consider both the metabolic and muscular effort provides a more comprehensive perception of the user’s needs; however, this was not proposed in previous human-in-the-loop controls.

For this optimization problem, we implemented a CMA-ES optimizer in the low-level board of SmartOs (green block of Figure 1). The CMA-ES is a derivative-free search algorithm, commonly used for black-box scenarios, such as HITL controls in lower limb exoskeletons [9,11]. It stochastically samples several candidates over several generations and learns by adapting the covariance matrix of a multivariate normal distribution in real time [32]. We computed the initial parameters of the algorithm by following the strategy proposed by Hansen et al. [32]. Table 1 presents the initial parameters.

An iteration (λ) refers to a tested set of control parameters. The optimizer performed a certain number of iterations before updating its internal parameters (called a generation). The CMA-ES optimization process was finished after performing 20 generations (each with 6 iterations). Thus, the optimization lasted for 20 min, similarly to in previous studies [10,12,13], as each iteration took 10 s to complete (the time required by the EGPR to estimate the metabolic cost).

During each CMA-ES iteration, the user’s physical exertion was evaluated through a cost function (CF) computed in real time using the weighted sum of two parameters, the estimated metabolic cost (MC) and the cumulative torque of the interaction torque (IT), as presented in Equation (Equation 1), where $\omega {\;}_{1}$ and $\omega {\;}_{2}$ are the weights of each parameter.
(1)$$CF=\omega {\;}_{1}MC+\omega {\;}_{2}MC$$

We tested various combinations of $\omega {\;}_{1}$ and $\omega {\;}_{2}$ by performing multiple empirical tests and, consequently, evaluating the CMA-ES performance and the user response. These tests showed that the optimized control parameters responded well to equal unitary weights. The minimization of these two physiological parameters in an equal manner was imperative for good CMA-ES performance, since the two signals have an analogous impact on the user’s exertion.

The control parameters adapted in real time by the CMA-ES were the torque magnitudes of flexion and extension peaks of the knee torque profile. Figure 3 presents the knee torque profile, which represents the torque trajectory that the exoskeleton must perform over one gait cycle. The profile is composed of three sections. While the right leg is fully stretched, the torque is null. The first positive torque curve causes the knee flexion motion, and the following negative curve executes the knee extension motion.

Figure 3 presents the possible torque profiles tested by the CMA-ES algorithm. The optimizer was set to test flexion torque peaks between 15 Nm and 20 Nm, and extension torque peaks between −15 Nm and −20 Nm. The torque profile was initialized with values of 17.5 Nm and −17.5 Nm for the flexion and extension peaks, respectively; intermediate values in the range of tested control parameters. These torques are values allowed by the SmartOs DC motor and can guarantee a comfortable gait for every user, something previously tested and empirically evaluated with various persons. Subsequently, the CMA-ES explored various combinations of these two parameters in each iteration, until the maximum number of generations was reached, identifying the optimal profile, i.e., the torque profile that minimized the SmartOs user’s physical effort.

### 2.4. Torque-Tracking Control

We developed a torque-tracking control as the low-level of the HITL control and implemented it in the low-level board of SmartOs (red block of Figure 1). The torque-tracking control aims to adjust the exoskeleton actuator’s torque based on the reference torque profile, as defined by the peak flexion and extension torques. The reference torque profile is adjusted in real time and individually for each subject during HITL optimization. We used a natural cubic spline interpolator algorithm to generate the reference torque profile based on the flexion and extension torque peaks optimized by the CMA-ES. First, the points that define the torque profile are identified. Then, the spline algorithm creates a piece-wise polynomial cubic curve that intersects the desired points by obtaining three arrays of constants (*a*, *b*, and *c*), based on the approach introduced in [33]. By estimating the gait cycle percentage (*P*), it is possible to determine the desired torque at each gait stage (T(P)) using Equation (Equation 2) [33],
(2)$$T\left(P\right)=t_{i}+a_{i}\left(P-p_{i}\right)+b_{i}{\left(P-p_{i}\right)}^{2}+c_{i}{\left(P-p_{i}\right)}^{3}$$
where i∈{1,2,…,n} and $p_{i}\leq P<p_{i+1}$, *n* is the number of points that define the torque profile, and *p* and *t* are the arrays defining these points coordinates, i.e., the gait cycle percentage and torque value, respectively. In this case, the torque profile was composed of 13 points (n=13), as required to correctly shape the curve according to our needs.

Every 10 milliseconds (the mid-level control period), a gait cycle phase modulator is used to estimate the gait cycle phase, defined in percentage (P). The modulator infers the duration of a gait cycle based on the selected gait speed and performs a linear increment of the gait phase percentage [34]. Then, this value is used to compute the desired torque by applying Equation (Equation 2). Lastly, the desired torque is sent to a proportional–integral–derivative (PID) torque controller, a closed-loop controller that minimizes the difference between the desired torque and the real motor torque, measured by the Hall effect sensors. Every millisecond, the PID controller transmits a command to the DC motor to force its torque to follow the reference pattern. The PID proportional, integral, and differential gains were optimized with the Ziegler–-Nichols method and their values were 135, 1.5, and 1.5, respectively.

## 3. Experimental Validation

The presented solution was experimentally validated according to two protocols with healthy volunteers. The first experiment enabled us to assess the performance of the EGPR model in estimating the participants’ metabolic cost, in real time, in comparison to the indirect calorimetry method. The second experimental protocol was intended to analyze the efficacy of the HITL control in reducing the exoskeleton users’ physical effort, comparatively to other control methods.

All participants were informed of the study’s objectives and methodology and received an informed consent form, which they read and signed. Both protocols were approved by the University of Minho Resear h in Life and Health Sciences Ethics Committee.

### 3.1. Metabolic Cost Estimation

#### 3.1.1. Participants

The EGPR performance for metabolic cost estimation was tested during a real-time experimental protocol with 5 volunteers (3 males and 2 females). The participants were healthy individuals without clinical history or evidence of motor and cognitive impairments. They had ages between 22 and 29 years old (25 ± 2.9 years), body masses ranging from 65 kg to 99 kg (77.8 ± 13.4 kg), and BMIs between 24.0 kg/m^2^ and 29.2 kg/m^2^ (26.8 ± 2.0 kg/m^2^).

#### 3.1.2. Instrumentation and Protocol

We instrumented the participants with four IMUs from the InertialLab system, to measure the chest, wrist, waist, and ankle 3D acceleration (as shown in Figure 4). Further, the participants wore a respirometer device (K5, Cosmed, Rome, Italy) to estimate the ground-truth metabolic cost by indirect calorimetry. The participants did not wear the exoskeleton during the experiment but were equipped with a backpack accommodating the SmartOs control system, which executed the EGPR model every 10 s.

The participants performed three activities: (i) standing; (ii) treadmill walking at 1.5 km/h, 2.0 km/h, and 3.0 km/h; and (iii) sitting in a regular chair without arms. Each task was performed for 10 min, without rest breaks, and the order of the three walking speeds was at random for each participant. This procedure replicated the experimental protocol performed by Ingraham et al. [18], whose data were used to train our regression model. Figure 4 presents a participant performing the experimental protocol while instrumented with the equipment used.

#### 3.1.3. Model Evaluation

The data collected during the protocol were composed of the metabolic cost estimated by the SmartOs system through the regression model, and the respirometer mask through indirect calorimetry, which provided the ground-truth. Similarly to the procedure followed during the EGPR training, the ground-truth was directly computed by the unprocessed breath-by-breath data obtained by the respirometer. We evaluated the EGPR model’s performance by computing the RMSE and MAPE between the EGPR estimation and the ground-truth. These metrics were computed using Equations (Equation 3) and (Equation 4), where *N* is the number of estimations, y(i) is the *i*th ground-truth metabolic cost, and $\hat{y}\left(i\right)$ the *i*th metabolic cost estimated by the EGPR model. Furthermore, we also analyzed the concordance between the two signals using Bland–Altman plots. The computational load of the regression model within the SmartOs architecture was also studied in bench tests.
(3)$$RMSE=\sqrt{\frac{\sum_{i=1}^{N}{\left(y\left(i\right)-\hat{y}\left(i\right)\right)}^{2}}{N}}$$
(4)$$MAPE=\frac{1}{N}\sum_{i=1}^{N}\left|\frac{y\left(i\right)-\hat{y}\left(i\right)}{y(i)}\right|$$

### 3.2. HITL Control

#### 3.2.1. Participants

We carried out an experimental validation to demonstrate the proof-of-concept of the proposed HITL control with one healthy participant. The participant was a 23-year-old female, with a body mass of 65 kg and a BMI of 24.5 kg/m^2^, a value within the range of BMIs of the participants of Ingraham’s study [18]. She had no previous experience in using a knee exoskeleton.

#### 3.2.2. Instrumentation and Protocol

During the experiment, the participant used the SmartOs knee exoskeleton on her right leg. She also wore the four IMUs required by the EGPR model to estimate the metabolic cost, placed on the chest, right wrist, left waist, and right ankle. Figure 5 depicts the participant performing the experimental protocol. A video showing the participant walking with the exoskeleton in HITL control is available in the Appendix A.

The participant walked on a treadmill at 1.5 km/h. The protocol was composed of three phases: (i) acclimatization to the device; (ii) online optimization of the torque profile performed by the CMA-ES optimizer for a maximum of 20 min; and (iii) comparison of the HITL control (i.e., optimal torque-tracking control) with other exoskeleton control strategies. The participant rested for 10 min between each phase to guarantee that no phase would affect the subsequent one [11].

Figure 6 presents the activities performed by the participant during each phase. Firstly, the participant underwent an acclimatization phase that lasted until she felt comfortable while walking with the device. In this phase, the participant walked with the device in three different control strategies: (i) zero-torque control, where the exoskeleton functioned in a transparent mode [35]; (ii) position-tracking control, a closed-loop control used to manipulate the actuator to follow a fixed reference joint angle trajectory [35]; and (iii) torque-tracking control with a fixed reference torque profile with flexion and extension torque peaks of 20 Nm and −20 Nm, respectively. Additionally, the participant rested between each mode until she felt capable of continuing the experiment.

In the protocol’s second phase, the participant walked with the exoskeleton until the CMA-ES optimizer had found the optimal torque profile, i.e., the torque that minimized her physical effort. During this phase, the optimizer generated new control parameters (torque peaks) every iteration (10 s), over 120 iterations. These were sent to the torque-tracking control, which computed a torque profile for each set of torque peaks and manipulated the knee exoskeleton to follow the desired torque trajectory. At each iteration, the CMA-ES analyzed the user’s response to the set of control parameters and determined the next values of peak torques to be tested. The process stopped when the peak torques that minimized her physical effort were found. After optimizing the assistance given to the participant, the CMA-ES was not executed again, and the HITL control applied the optimal peak extension and flexion torques to the torque profile, which was used by the torque-tracking control to manipulate the exoskeleton.

The third phase aimed to evaluate the efficacy of the optimal torque profile in reducing the user’s effort, and to benchmark the HITL control with commonly applied control strategies. The participant started the third phase by performing a trial without the exoskeleton. Then, she walked with the exoskeleton in three assistive strategies: (i) zero-torque control; (ii) torque-tracking control with a fixed reference torque profile with maximum extension and flexion peaks (20 Nm and −20 Nm, respectively); and (iii) HITL control, i.e., torque-tracking control with the optimal torque profile obtained by the CMA-ES optimizer in the second phase. This experiment was based on protocols performed in the literature [12,13,14].

#### 3.2.3. Control Evaluation

To evaluate the HITL control effectiveness, we compared the user’s metabolic cost and interaction torque during the four tested conditions and analyzed if using the HITL control led to an increase or decrease in the user’s physical effort. Boxplots were generated to display the variation in the participant’s metabolic cost and interaction torque during all conditions. Additionally, we also analyzed the variation in the participant’s physical effort and the two control parameters over the CMA-ES iterations.

## 4. Results

This section presents the results obtained during the experimental validation of the metabolic cost estimation and the HITL control.

### 4.1. Metabolic Cost Estimation

In a preliminary phase, the regression model was trained and validated using the leave-one-subject-out cross-validation method. In this offline validation, the model obtained an RMSE of 0.31 W/kg. One participant from the dataset [18] was randomly chosen for the offline test with unseen data. During this offline test, the EGPR model obtained an RMSE of 0.45 W/kg.

Then, the model was tested in real time by involving 5 participants in an experimental protocol, where EGPR estimation was performed simultaneously with the indirect calorimetry method. Figure 7 depicts the evolution of the estimated metabolic cost by the EGPR model and indirect calorimetry during the experiment for a representative participant. We observed that, in general, the EGPR model slightly underestimated the metabolic cost in comparison to the indirect calorimetry method during walking assistance. This finding was consistent across all participants. The results also demonstrate that the EGPR model presented a less noisy signal than indirect calorimetry.

We compared the metabolic cost estimation provided by the EGPR model and the indirect calorimetry method for the 5 volunteers that participated in the experimental validation, by computing the RMSE and MAPE between the two signals. Figure 8 presents the average RMSE and MAPE for each motor activity, as well as the lowest and highest values of these errors across the 5 participants. On average, the RMSE and MAPE values were 0.66 W/kg and 26%, respectively, across all activities and participants.

Figure 9 presents Bland–Altman plots of the data for each activity, comparing the estimations made by the EGPR model (“Estimation”) and the indirect calorimetry method (“Target”). Each plot shows the agreement between the two metabolic cost estimations, identifying estimation biases, i.e., the average value of the difference between the ground-truth and the EGPR estimation. The bias increased with the exigence of the exercise, being, therefore, higher for walking at 3 km/h and lower for the sitting activity. Figure 9 also shows that the difference between the two estimations was less dispersed for the static activities and walking at 3 km/h, a speed closer to a natural gait.

Additionally, we assessed the computational load of the pipeline of the regression model implemented in the CCU of SmartOs (Figure 2). The CCU took, on average, 0.23 ms to read and save the acceleration data, which were measured every 10 ms. Furthermore, to process the data and estimate the metabolic cost every 10 s, the model took 2.6 and 9.9 ms, respectively. Therefore, all computational processes were executed within the set requirements.

### 4.2. HITL Control

The results from the experimental validation of the HITL control strategy were analyzed to assess the behavior of the optimization algorithm during the HITL optimization phase and the performance of the optimal torque-tracking control when compared to other control strategies.

Figure 10 presents the progression of the real-time torque optimization with the CMA-ES algorithm. It shows the variation in the flexion and extension torque peak values (i.e., the control parameters) and the cost function value (i.e., the user’s physical effort) over the 120 iterations. After going through the optimization process, which lasted 20 min, the CMA-ES considered as optimal control parameters flexion and extension torque magnitudes of 18 Nm and −17.9 Nm, respectively.

Figure 11 depicts boxplots of the user’s estimated metabolic cost variation throughout the four tested conditions in the third phase of the experimental protocol. Furthermore, Figure 11 presents the percentage reduction in the metabolic cost between the zero-torque control and the HITL control (i.e., the optimal torque control). The metabolic cost was higher with the zero-torque and the fixed torque controls, and lower with the optimized HITL control. The HITL strategy enabled a 7.3% reduction in metabolic cost compared to the zero-torque control.

Figure 12 presents the median of the module of the interaction torque during three different control strategies in the third phase of the experimental validation. Additionally, Figure 12 shows the reduction in the interaction torque when comparing the zero-torque control and HITL control (i.e., the optimal torque control). The interaction torque measured when the participant experienced the optimized HITL control was similar to the fixed torque control and 32.3% lower than the zero-torque control.

## 5. Discussion

The main goal of this work was the development of a novel HITL control for a wearable knee exoskeleton, capable of finding the user-specific optimal assistance that minimizes the users’ physical effort while they walk with the exoskeleton. The control presented in this study innovated by adapting the exoskeleton assistance considering two indicators of physical effort, the metabolic cost and the interaction torque, representing the user’s metabolic and muscular effort, respectively. Additionally, this was the first HITL control strategy that relied on the metabolic cost estimated by a machine-learning model; thus, fostering a more practically assistive approach than indirect calorimetry for real use in industry. It should be noted that walking was the activity that required the higher metabolic effort from the participants, roughly two- to three-times higher when compared to the sitting and standing activities. Therefore, due to these results, this work focused on assisting exoskeleton users when walking.

### 5.1. Metabolic Cost Estimation

Similarly to this work, various previous studies have developed machine- and deep-learning regression models to estimate metabolic cost based on data from inertial sensors. Slade et al. [22] used as inputs the shank and thigh acceleration and obtained a MAPE of 13.7%. Other studies fused acceleration data with additional inputs, such as the heart rate and muscle activity (RMSE of 0.36 W/kg) [24]. The results obtained across the literature are highly variable, as they are dependent on the inputs used, the type of regression models, and the activities performed. Despite that, there is evidence that the RMSE is typically higher for more demanding exercises [22,23]. This is supported by our findings, as the average RMSE of the model was lower for the static tasks and increased with walking speed, as seen in Figure 8. The higher RMSE for the walking activity may be explained by the higher intra- and inter-subject variability in more demanding exercises. The Bland–Altman plots depicted in Figure 9 also show an increase in the difference between the EGPR estimation and the ground-truth with the physical exigence of the task.

### 5.2. HITL Control

#### 5.2.1. HITL Optimization

CMA-ES optimizers have been widely used across the literature to adapt the assistance of lower limb exoskeletons in real time [9,10,11,16,17]. The time it takes to fully optimize an exoskeleton’s assistance is directly dependent on the number of control parameters adapted but is also dependent on the termination conditions enforced for the CMA-ES algorithm. The literature studies that used a CMA-ES optimizer took from 12 min/parameter [16] to 36 min/parameter [17]. The most common control parameters optimized in the literature are two [8,10,16] or four peaks [9,11,17] of the torque profile of an ankle exoskeleton.

Considering these literature outcomes, we also used a CMA-ES optimizer. The behavior of our CMA-ES algorithm during the optimization showed that the values of the control parameters (the flexion and extension torque peaks) fluctuated significantly at the beginning of the optimization process, and converged to their optimal final values over time, as the optimizer became smarter and closer to finding the optimal solution. The value of the objective function (the user’s physical effort) presented a similar progression, getting closer to its minimum as the CMA-ES was learning. The algorithm took a total of 20 min to optimize the two control parameters of the knee exoskeleton, requiring less than 12 min/parameter to find the best exoskeleton assistance.

Additionally, the optimal flexion and extension peaks had approximately equal absolute values (≈18 Nm). This optimal value corresponds to an intermediate value between the maximum and minimum possible peaks (20 Nm and 15 Nm, respectively). Higher torque peaks resulted in faster and sudden movements, which increased the user’s metabolic cost, whereas lower torque peaks led to higher interaction torque between the user and the exoskeleton. Therefore, the optimizer settled for flexion and extension peaks that were not too high nor too low. Despite that, it is worth noting that the optimal torque profile is a user-oriented control parameter.

#### 5.2.2. Control Comparison

In this study, we developed a HITL control for a knee exoskeleton that aims to adapt its walking assistance, in real time, for each user, providing them with a personalized torque-tracking control. This is possible by employing an optimization algorithm that finds the control parameters that minimize the users’ physical effort, while they walk on a treadmill with the device. HITL controls have been increasingly studied in the literature, especially for lower limb exoskeletons for gait assistance. Regarding assistive effects, single-joint exoskeletons with HITL controls were able to reduce the metabolic cost of their users when compared to walking without the devices, achieving reductions that range from 7% [8] to 17.4% [12]. Similarly, our study verified that the user’s metabolic cost was 5.9% lower when using the exoskeleton with optimal torque control than when walking without the exoskeleton. These outcomes suggest that the exoskeletons bring positive and augmentative assistive advantages, even for healthy users, while performing a common activity, such as level walking.

Compared to walking with an exoskeleton in a zero-torque control, state-of-the-art HITL controls achieved metabolic cost reductions of 17.4% [12] and 24.2% [11] when using a single-joint ankle exoskeleton and 48% when using a hip-knee-joint exoskeleton [36]. Our results showed that the user achieved the lowest metabolic cost when walking with the exoskeleton set for user-specific optimal torque control, suggesting that the HITL strategy was effective in minimizing this physiological parameter. Compared to wearing the exoskeleton in a zero-torque control, the HITL control reduced the user’s median metabolic cost by 7.3%. This effect is in accordance with the findings of Zhang et al. [9] for slow walking, who showed that the percentage reduction in metabolic cost depends on the gait speed, reporting 3%, 33%, and 39% for slow (2.7 km/h), normal (4.5 km/h ), and fast walking (6.3 km/h), respectively, when comparing a HITL control with walking in a zero-torque mode [9]. Moreover, the HITL control also led to a reduction of 32.3% in the user–exoskeleton median interaction torque when compared to the zero-torque control; thus, it appeared that the control was able to minimize the user’s muscular effort. Overall, these findings highlight the assistive advantages of active HITL assistance over transparent assistance (zero-torque control), regarding the users’ metabolic and muscular effort. When comparing a torque-tracking control with torque peaks fixed to the maximum assistive values of the exoskeleton with the optimal peak values found by the HITL optimization, we demonstrated that the HITL control led to a 7.8% reduction in the user’s metabolic cost while maintaining similar user–exoskeleton interaction torque values. This showed that the fixed maximum torque-tracking control, despite enabling low interaction torques, resulted in accelerated movements that increased the user’s metabolic cost. Thus, the conducted use-case validation highlighted the relevance of setting user-specific control parameters in line with the current paradigm of personalized assistance, rather than using fixed references across subjects.

### 5.3. Limitations

The work presented here served as a proof-of-concept of a HITL control that optimizes both the user’s metabolic cost estimated by a machine-learning regression model and the human–robot interaction torque, a metric related to the user’s muscle activity. The results favor the future implementation of HITL control in active exoskeletons over zero-torque and non-user-specific assistance. However, one of the limitations of this work was the reduced sample size used to validate the EGPR model’s performance during the real-time metabolic cost estimation. We aim to perform a more extensive data acquisition to further validate this regression model. Another limitation was that the HITL control was tested with a single participant walking at a slow speed. In the future, we expect to adapt the HITL control to assist further users with more physically demanding activities, such as load carrying and lifting.

## 6. Conclusions

This study presented a HITL control that appeared to reduce the users’ physical effort while walking with a knee exoskeleton, measured by their interaction torque with the device and metabolic cost. The control parameter optimization was performed by a CMA-ES algorithm, in real time. The proposed strategy goes beyond the state of the art by estimating the user’s metabolic cost with a machine-learning regression model, based on data obtained from a minimal number of input signals measured using wearable accelerometers and the user’s BMI. The regression model achieved less noisy metabolic cost estimates and enabled a reduction in the time required to estimate the steady-state metabolic cost from 3 min to 10 s, compared to the indirect calorimetry method.

Furthermore, to the authors’ best knowledge, this was the first HITL control that simultaneously minimized metabolic and muscular physiological parameters. The results suggested that walking with the developed HITL control led to a reduction in the user’s metabolic cost when compared to walking without the device and with the device in a zero-torque and non-user-specific torque control. The user’s muscular effort also decreased when assisted using the torque-tracking control with non- and user-specific parameters, compared to the zero-torque control.

## Figures and Tables

**Figure 1 sensors-24-03305-f001:**
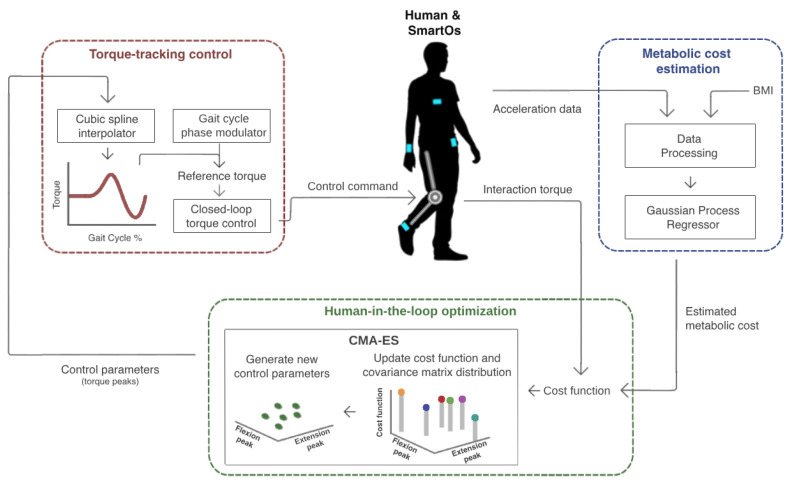
HITL control proposed to minimize an exoskeleton user’s physical effort assessed in terms of metabolic cost and human–robot interaction torque.

**Figure 2 sensors-24-03305-f002:**
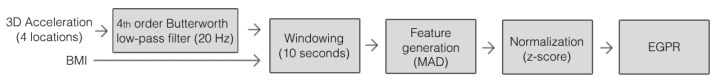
Method for processing input data to estimate metabolic cost.

**Figure 3 sensors-24-03305-f003:**
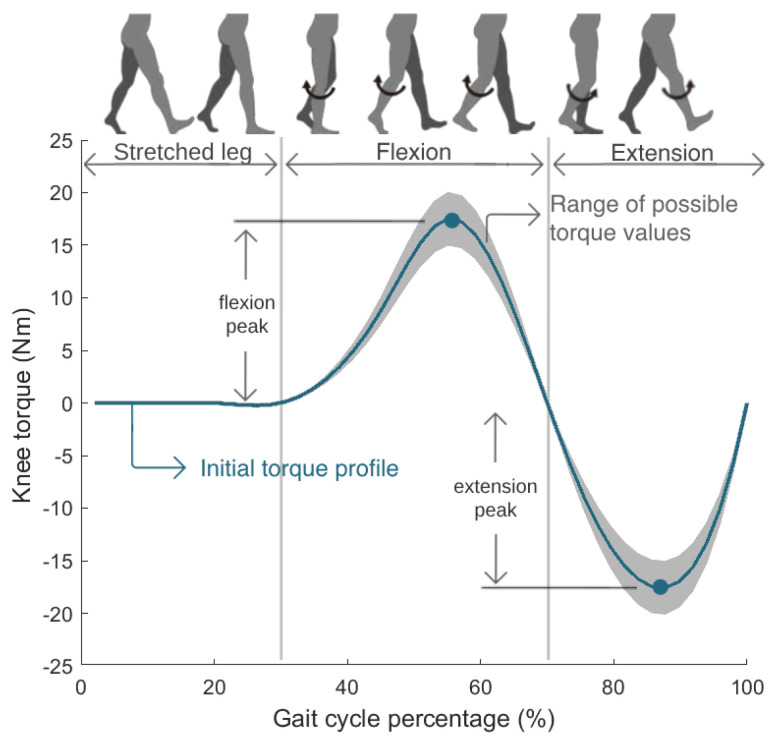
Knee torque profile optimized in real time by the CMA-ES.

**Figure 4 sensors-24-03305-f004:**
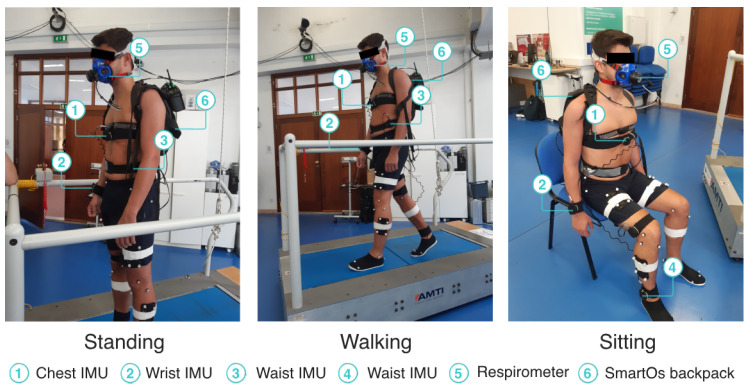
Participant performing the standing, walking, and sitting activities.

**Figure 5 sensors-24-03305-f005:**
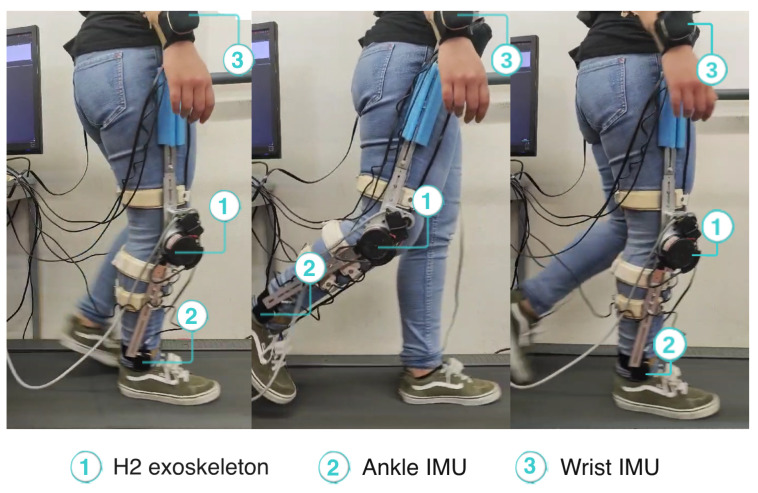
Participant walking with the exoskeleton during HITL validation.

**Figure 6 sensors-24-03305-f006:**
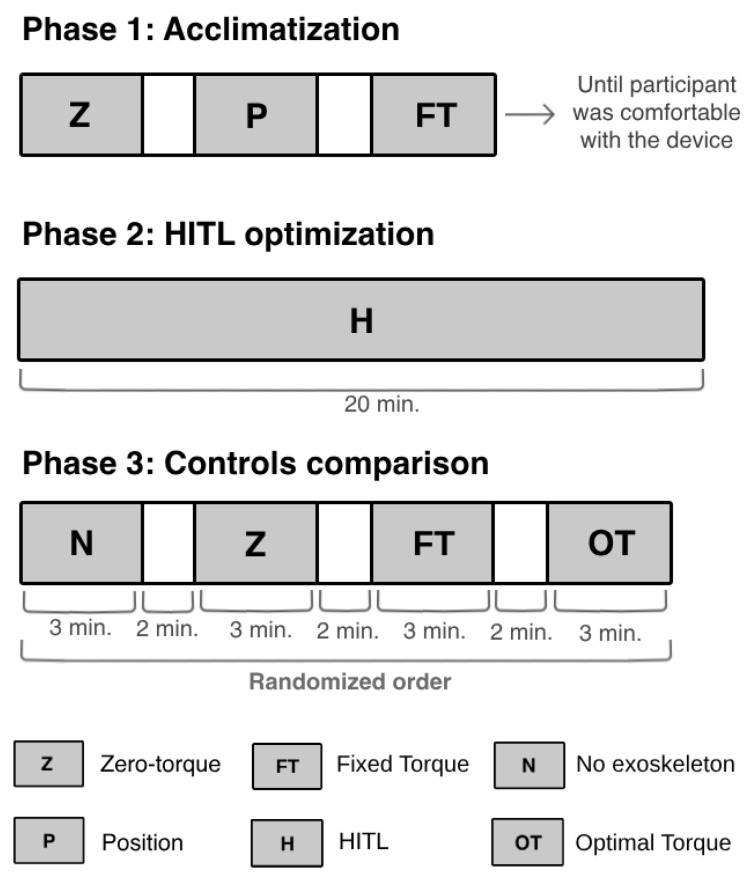
Experimental protocol performed to validate the HITL control.

**Figure 7 sensors-24-03305-f007:**
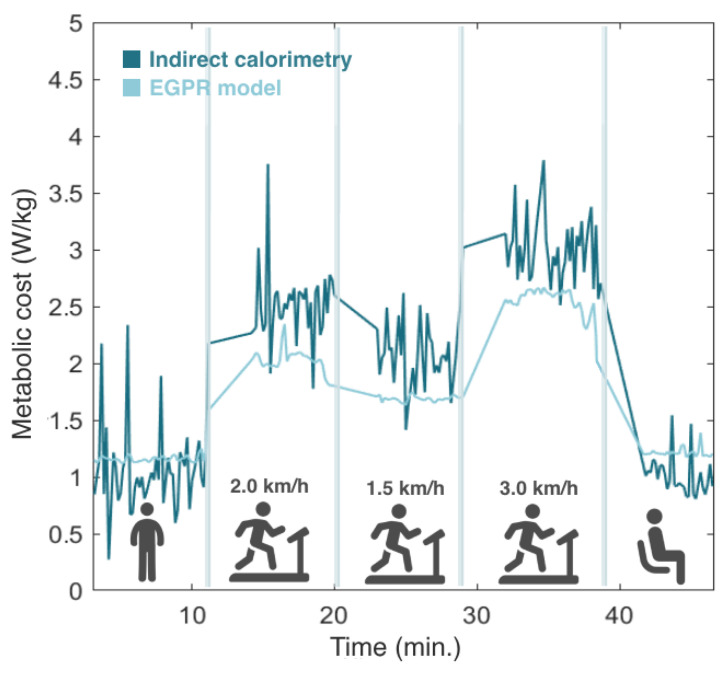
Comparison of the metabolic cost estimated by indirect calorimetry and the regression model. Results from one representative participant.

**Figure 8 sensors-24-03305-f008:**
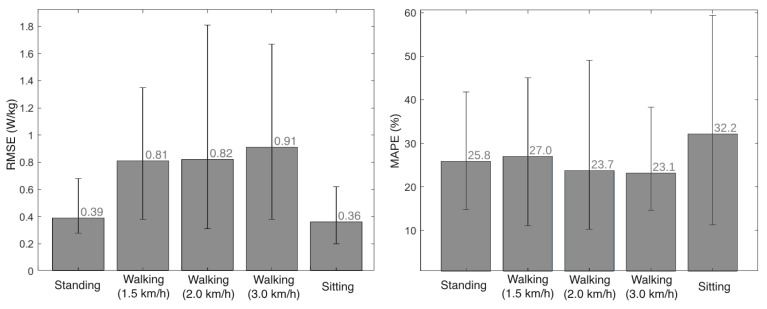
Mean and variation of the RMSE (**left view**) and the MAPE (**right view**) between the metabolic cost estimated by indirect calorimetry and the regression model, for each activity.

**Figure 9 sensors-24-03305-f009:**
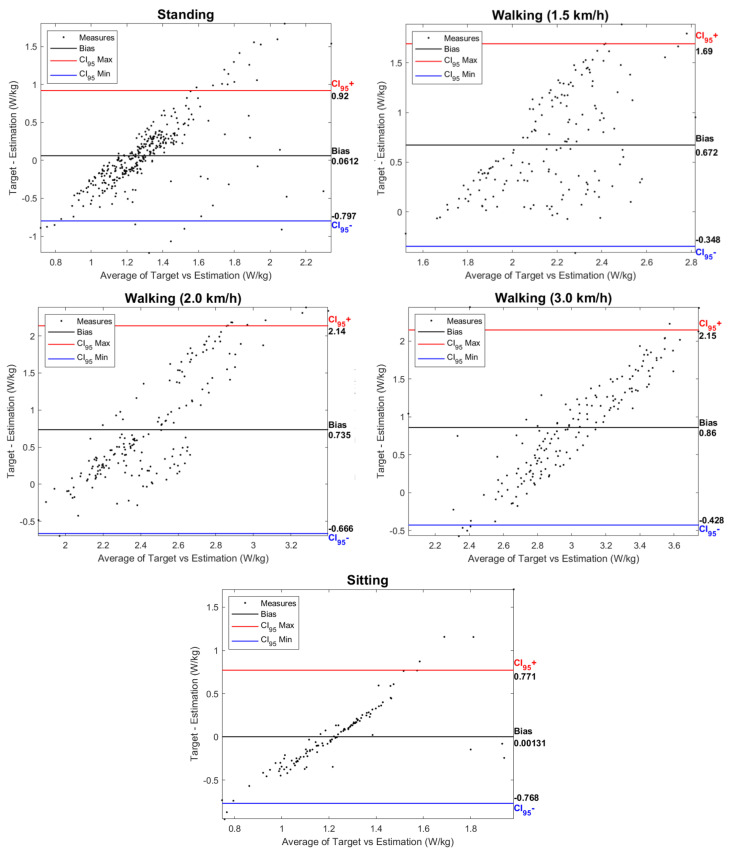
Bland-Altman plots for each activity with data from the 5 participants. Indication of the bias between the ground-truth and the estimated metabolic cost. Each plot also depicts the upper and lower limits of agreement, with red and blue lines, respectively.

**Figure 10 sensors-24-03305-f010:**
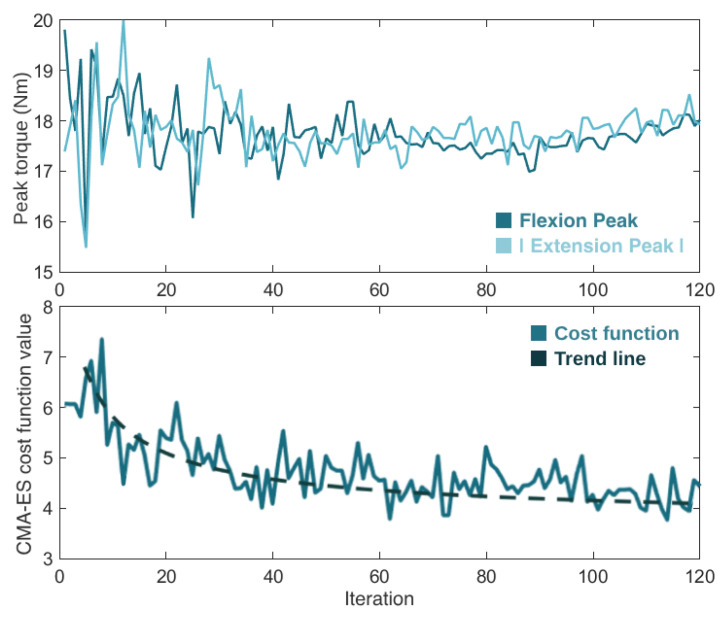
Evolution of the torque flexion and extension peaks of torque (**top view**) and the cost function value (**bottom view**) during the 120 iterations (20 min) of the CMA-ES optimizer.

**Figure 11 sensors-24-03305-f011:**
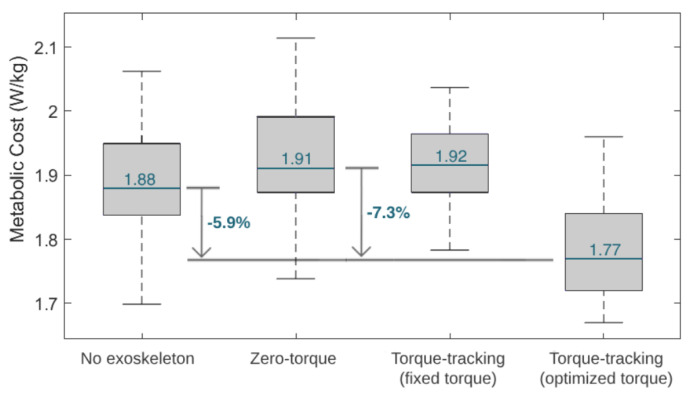
Boxplots displaying the metabolic cost variation during the tested conditions. The median value during the tested conditions is marked in blue.

**Figure 12 sensors-24-03305-f012:**
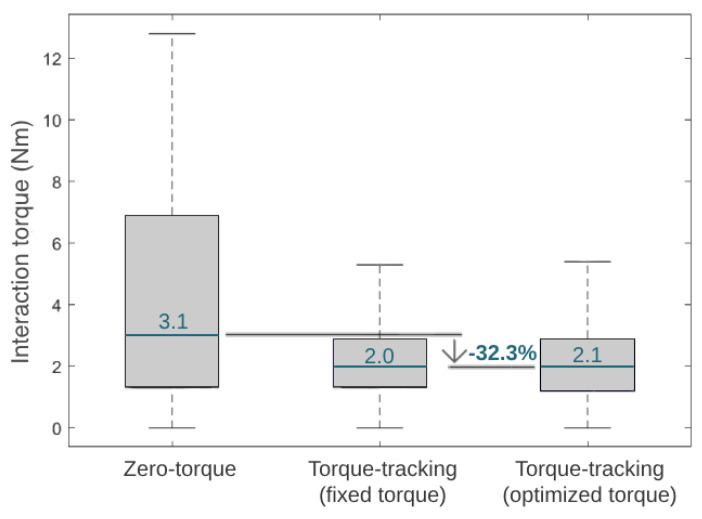
Boxplots displaying the interaction torque variation and median value during the tested conditions.

**Table 1 sensors-24-03305-t001:** CMA-ES initial parameters.

*n*	2	σ	1.4	$\omega_{1}$	0.59	$C_{\sigma }$	0.51
λ	6	$\sigma_{1}$	1.4	$\omega_{2}$	0.29	$C_{c}$	0.51
$d_{\sigma }$	1.51	$\sigma_{2}$	1.4	$\omega_{3}$	0.12	$\mu_{\mathrm{cov}}$	2.24
$c_{\mathrm{cov}}$	0.12	C	I	μ	3	$\mu_{\mathrm{eff}}$	2.24

## Data Availability

The raw data supporting the conclusions of this article will be made available by the authors on request.

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
