# Peer review of "Human-in-the-Loop Optimization of Knee Exoskeleton Assistance for Minimizing User’s Metabolic and Muscular Effort"

_sensors, 2024, doi:10.3390/s24113305_

Round 1
Reviewer 1 Report
Comments and Suggestions for Authors
Dear authors, I congratulate you for the work done. However, in order to improve the quality of the article, the following comments are included to be made.
In general, relevant information is mentioned that refers to previous works. However, it would be important for the document to include these aspects in general terms, making the reading and understanding of the information presented more readable.
At Metabolic Cost Estimation, page 4-5, a filter is considered within the process. Could the authors include a paragraph explaining the purpose of it? why is it considered with 4 order? Is there any implication of it being different in order?
Regarding the optimization process at page 5, it is mentioned that a cost function is used to perform this procedure. It will be better for the reader if the authors could explain more about it. Could the authors present the cost function definition? The correspondet explanation should be also included.
At page 7, could the authors give an insight of the gait cycle phase estimator? How is it working?
When evaluating results at page 13, particularly with respect to PID control, one typically expects to see that the operation of the control ensures compliance with the control objectives. Based on the presented results, this is done based on the effort graphs included in the manuscript. However, from this reviewer's point of view, it would be important to present results of the process of minimizing the difference between the desired and actual bending value. Could the authors present a more in-depth explanation of this? On what basis is the control said to perform the task for which it was designed?
Due to the nature of the addresed issue, it is proposed to include https://doi.org/10.3390/act10070166 as reference... to the author's consideration.
Again, congratulations on the results presented throughout this paper.
Comments on the Quality of English Language
Minor details should be fixed.
Author Response
Thank you very much for taking the time to review this manuscript. Please find the detailed responses in the following attachment and the corresponding revisions/corrections highlighted in the re-submitted manuscript

Reviewer 2 Report
Comments and Suggestions for Authors
1. I'm surprised to learn that regression models can accurately estimate physiological signals for the first time in the literature, as presented in this paper. This sounds unrealistic, as regression models are probably among the first choices for dealing with time-series data such as physiological signals. Please revise or double-check the literature.
2. It is not surprising that regression models performed well in this work. This is common from the data science perspective, as this is precisely what regression models should do. The challenge or question is probably that the authors tested their models on only one subject. Whether this model provides sufficient capabilities in generalization is questionable. More data and tests are needed to make the conclusions drawn in this work.
3. The abstract, introduction, and conclusion sections lack information. Although the paper title states that the authors' research focus is HITL, the implementation method is the CMA-ES algorithm. None of that is mentioned anywhere in the abstract, introduction, or conclusion sections. This is highly misleading and must be revised.
4. The authors must also discuss in detail the limitations of this work.
5. The exoskeleton used seems to be a specific product, which lacks the necessary information to understand this work better.
6. It seems that HITL control is a better terminology for implementing CMA-ES algorithms, as demonstrated in Figure 1. Please clarify the changes made to the manuscript. Also, it is essential to include a video demonstration of the effectiveness of HITL control.
Author Response

(The authors gave the same response as above.)

Reviewer 3 Report
Comments and Suggestions for Authors
Thank you for the opportunity to review the manuscript titled "Human-In-The-Loop Optimization Of Knee Exoskeleton Assistance For Minimizing User’s Metabolic And Muscular Effort," in which the authors describe a HITL control strategy for a knee exoskeleton intended to minimize the users’ muscular effort and metabolic cost.
The manuscript is relatively well written and informative; however, I do have some major concerns and minor, proofreading-related comments that should be addressed for the manuscript to be eligible for publication. Please see below.
MAJOR ISSUES
1. Abstract: The sample sizes for both experiments should be provided.
2. In Lines 15, 16, and 106, the term "assisted working" is used, and in Lines 86-88, the authors state that their HITL control was "designed for an industrial exoskeleton" to assist workers "while carrying and lifting heavy loads." Yet, the control was only tested during unloaded standing, sitting, and walking. This should be justified; alternatively, the term "assisted working" should be replaced with "assisted walking" and the statement in Lines 86-88 revised.
3. Metabolic Cost Estimation
3.1 The sample size for metabolic cost estimation (n = 5) was considerably smaller than that in the referenced similar studies (especially [17], [21], [23], and [36]), which may be the reason for the large variation of RMSE results (sitting: 0.2-0.6 W/kg; standing: 0.3-0.7 W/kg; walking 1.5 km/h: 0.4-1.3 W/kg; walking 2.0 km/h: 0.3-1.8 W/kg; walking 3.0 km/h: 0.4-1.7 W/kg); yet, this limitation is not addressed in the manuscript. The chosen sample size should be justified.
3.2 Lines 337-339: "We observe that, in general, the EGPR model slightly underestimates the metabolic cost in comparison to the indirect calorimetry method." Based on Figure 7, this is only true for walking; during standing and sitting, the model actually appears to have overestimated metabolic cost.
3.3 1 kcal/min equals 70 W; thus, if my understanding is correct, the RMSE values reported in kcal/min by Lucena et al. [17], Sazonov et al. [21], Ni et al. [23], and Strath et al. [36] cannot be compared to the values reported in W/kg by Ingraham et al. [16], Lopes et al. [22], and the present authors. Considering that the authors only report RMSE in W/kg, comparing their results to those expressed in kcal/min (Lines 402-408) seems meaningless and could mislead a careless reader.
3.4 The results obtained by Lopes et al. appear to be far superior (RMSE = 0.36 W/kg) to those reported by the authors. What are the advantages of the model proposed in the present manuscript?
4. Model Evaluation
4.1 Model evaluation based on data obtained from a single participant cannot be reliable. The chosen sample size should be justified and addressed in the Limitations.
4.2 Figures 11 and 12: If my understanding is correct, "zero-torque control" refers to the situation where the participant is wearing the exoskeleton, but the exoskeleton is not activated and does not provide any assistance (i.e., it is in "slack" mode; this is different from "passive assistance," as stated in Lines 472-473). Such a scenario is highly unlikely in real-life settings, as the exoskeleton only adds weight to the lower limb, which increases metabolic cost. Activating the exoskeleton, therefore, understandably, leads to a larger metabolic cost reduction when compared to walking in a zero-torque condition than it does when compared to walking without the device (as also suggested by reference [9] in Line 43).
Although previous studies have made similar comparisons, comparing the HITL control to zero-torque control is of limited relevance and can be practically misleading, as it is unlikely that people would opt to wear an inactive exoskeleton when it is easier to move without the device. When compared to walking without the exoskeleton, the HITL strategy actually only resulted in a 5.9% reduction in metabolic cost (Figure 11), and produced a 0.1-Nm larger interaction torque than the "fixed torque control" strategy (Figure 12). The first fact is correctly stated in Line 454; however, this is not mentioned in the Abstract. I suggest a revision of the figures and associated text.
Comments on the Quality of English Language
PROOFREADING ISSUES
1. There are several instances of excessive use of "the," for example:
- Lines 78-80: "... the input signals that have resulted in better metabolic cost estimation are: (i) the waist, wrist, and ankle acceleration; (ii) THE muscle activity; (iii) THE heart rate; (iv) THE breath frequency; and (v) THE minute ventilation ..."
- Line 150: "The data selected from the dataset included THE acceleration measurements at the ..."
- Line 154: "... only the data from THE relevant industrial activities, namely the standing, sitting, and walking activities ..."
- Figure 2.: "Method for processing THE input data to estimate THE metabolic cost."
- Line 177: "Combining physiological measurements that regard both the metabolic and THE muscular effort ... "
- Line 193: "... are the torque magnitudes of THE flexion and extension peaks of the knee torque profile."
- Line 226: "... based on THE gait speed ..."
- Figure 7.: "Comparison of the metabolic cost estimated by THE indirect calorimetry ..."
- etc.
2. Line 23: "... experienced muscular pains ..." -> "... experienced muscular pain ..."
3. Line 70: "breath rate" -> "respiratory rate" or "breathing rate"
4. Line 181: Please use either "lower-limb exoskeletons" or "lower limb exoskeletons" consistently throughout the manuscript.
5. Line 248: "They presented ages (...), body masses (...), and BMIs ..." -> The use of "presented" is awkward, please revise.
6. Line 312: "After optimizing the assistance to the participant the CMA-ES was not again executed ..." -> "After optimizing the assistance to the participant, the CMA-ES was not again executed ..."
7. Line 343: "... indirect calorimetry method, for the 5 participants ..." -> "... indirect calorimetry method for the 5 participants ..."
8. Line 355: "... a speed closer to a natural gait." -> "... a speed closer to natural gait."
9. Lines 381-382: "Additionally, it is displayed the reduction ..." - This is awkwardly stated, please revise.
10. Line 396: "... 2 to 3 times higher ..." -> "... 2- to 3-times higher ..."
11. Lines 424-425: "Regarding the control parameters optimized by the literature, these studies ..." - This is awkwardly stated, please revise.
12. Line 431: "... closer to find the optimal solution." -> "... closer to finding the optimal solution."
13. Line 455: "... the exoskeletons brings ..." -> "... the exoskeletons bring ..."
Author Response

(The authors gave the same response as above.)

Round 2
Reviewer 2 Report
Comments and Suggestions for Authors
No further comment. Very nice research.
Reviewer 3 Report
Comments and Suggestions for Authors
I would like to thank the Authors for thoroughly addressing my concerns.
The manuscript is now substantially improved and eligible for publication.